A lipidomic study on the lens epithelial cells of patients with age related cataracts

Gong Yingying
Wei Qingquan
Luo Liying
Qiu Wei
Jiang Yanyun JYY3378@shtrhospital.com
Shanghai Jiaotong University School of Medicine, Tongren Hospital , Shanghai , China
Redondo Beatriz
Electronic publication date: 2024 Sep 6
Publication date: 2024
Volume: 12
Electronic Location ID: e17998
Received 2023 Oct 5; Accepted 2024 Aug 7
Copyright: © 2024 Gong et al.
Copyright year: 2024
Copyright holder: Gong et al.
License: This is an open access article distributed under the terms of the Creative Commons Attribution License, which permits unrestricted use, distribution, reproduction and adaptation in any medium and for any purpose provided that it is properly attributed. For attribution, the original author(s), title, publication source (PeerJ) and either DOI or URL of the article must be cited.
License URL: https://creativecommons.org/licenses/by/4.0/

Keywords: Age related cataracts, Metabolomics, Lens epithelial cells, Liquid chromatography‑mass spectrometry

Funding: The authors received no funding for this work.

==============================
Age related cataracts (ARC) represent the main reason for blindness globally. The lens epithelial cells (LECs) participate not only in the metabolism of many substances in the lens but also in maintaining lens transparency. This study used lipidomics to investigate the metabolic differences in LECs of ARC patients with different severity, aiming at identifying potential metabolic biomarkers of ARC. Patients diagnosed with ARC and underwent cataract surgery at Shanghai Tongren Hospital were selected to participate in this study, which were classified as mild ARC group and severe ARC group. During their cataract surgery, anterior lens capsules(LCs) containing LECs were obtained. The lipidomics of LECs were analyzed using the liquid chromatography‑mass spectrometry (LC-MS). Potential pathways of lipids were searched for using databases such as the Kyoto Encyclopedia of Genes and Genomes (KEGG) and MetaboAnalyst platform. In LEC lipids, 26 lipids have been identified as potential biomarkers between mild ARC and severe ARC, with AUC values of 0.67–0.94. The pathway analysis results revealed that the Glycerophospholipid (GPL) metabolism was significantly influenced, indicating that these metabolic markers contribute significantly to regulating this pathway. The LEC metabolic spectrum demonstrates a proficient ability to differentiate between patients with varying levels of cataracts. Herein, we have successfully identified potential metabolic biomarkers and pathways that have proven to be valuable in enhancing our understanding of ARC pathogenesis. The finding has translational value for developing new cataract treatment methods in the future.

Introduction

Cataracts, a condition characterized by the opacification of the lens in the eye resulting in impaired visual acuity, is considered to be the main reason for blindness on a global scale (Congdon et al., 2004; Wang et al., 2016). This issue continues to be a significant public health concern in both developed and developing countries within the field of ophthalmology. Cataracts are categorized into various types based on their underlying causes, including age-related cataracts (ARC), traumatic cataracts, radiation cataracts, pediatric cataracts, and secondary cataracts. ARC represents the most prevalent type of cataract in adults (Liu et al., 2017). ARC leads to decreased vision, visual changes, and even blindness, which seriously affects the quality of life of patients (GBD 2019 Blindness and Vision Impairment Collaborators & Vision Loss Expert Group of the Global Burden of Disease Study, 2021). The prevalence of ARC in Chinese people aged 60 to 89 years is as high as 80% (Song et al., 2018). According to the seventh national census data in China, there are 260 million people aged 60 and above (https://www.stats.gov.cn/sj/tjgb/rkpcgb/qgrkpcgb/202302/t20230206_1902005.html; Major figures on 2020 population census of China). Based on this estimate, the number of cataract patients in China may have reached 208 million. As the aging population increases in most cities in China, the number of patients with cataracts has increased significantly, and many patients have even become blind.

The main pathological feature of cataract is lens opacity. The lens is a non-vascular, transparent structure wrapped by the lens capsule, whose function is to converge light to the retina (Liu et al., 2017). The lens comprises three main components: the LC (lens capsules), the lens epithelium, and lens fibers (He, Wang & Huang, 2017). The human lens epithelial cells (LECs) are a single layer of cuboidal epithelial cells that adhere closely to the inner surface of the anterior LC. The LECs primarily assume the role of governing the physiological balance of the lens, which includes the transportation of electrolytes and fluids. In addition, LECs can synthesize lipids to maintain the long-term integrity of the crystallin network within the lens, thereby maintaining the lens transparency (Bloemendal et al., 2004; Hsueh et al., 2022). Cataracts can be attributed to various risk factors, including oxidative stress, ultraviolet radiation, aging, and hyperglycemia. These factors have the potential to interfere with LECs and contribute to cataract development. LEC apoptosis was verified to serve as the shared cytological foundation for all cataract types, excluding congenital cataracts (Chua et al., 2017; Shu, Wojciechowski & Lovicu, 2017). The abnormal proliferation and differentiation of LECs exhibit a close association with the occurrence of cataracts and posterior capsule opacification (Meng et al., 2013). Recent research has shown that metabolic abnormalities in LECs can lead to the development of cataracts (Liu et al., 2023; Zhao et al., 2015). In mature lenses, LECs under the anterior capsule and equatorial region maintain lens homeostasis through material synthesis and exchange, slow proliferation and differentiation. A potential difference exists between LEC cell membranes and lens fiber cells in distinct regions, forming lens microcirculation, transporting energy and oxygen, as well as other substances. Abnormalities in these pathways can lead to the development of cataracts (Liu et al., 2023). However, the characteristics of lipids in LECs and the mechanisms by which they affect the maturity and development of cataracts still need to be elucidated.

Lipids are integral components in a multitude of biological processes occurring within human cells, including cellular signaling, the transfer of energy, and communication between cells. Lipids are involved to a great extent in governing these processes, thereby ensuring their efficient performance. Lipidomics is a new generation of omics technology driven by systems biology. It can perform quantitative analysis on all lipids in organisms. And it can search for the relative relationship between lipids and physiological and pathological changes (Rinschen et al., 2019). We can understand different biological chemical process and biological events from the metabolic perspective, such as the occurrence and diagnosis of diseases (Holmes, Wilson & Nicholson, 2008). In recent years, with the continuous development of lipidomics, our understanding has also been broadened. Lipidomics analysis is no longer a simple biomarker recognition tool but a new technology that can explore the driving factors of activity in physiological and pathological processes (Piazza et al., 2018; Rinschen et al., 2019).

Lipidomics has proven to be an effective approach in the research of eye disorders, enabling the identification of metabolic characteristics of various eye diseases. Several lipidomics studies using plasma or aqueous humor or urine have found differential lipids in identifying glaucoma, age-related macular degeneration, and diabetes retinopathy (Buisset et al., 2019; Chen et al., 2016; Lains et al., 2019; Lains et al., 2018; Pan et al., 2020; Xuan et al., 2020). In the study of the lens, there was a significant difference in the metabolomic compositions of the normal and ARC lens: almost all metabolite concentrations in the normal lens were found to be higher in comparison to the cataractous one (Tsentalovich et al., 2015). This indicated that ARC occurrence and progression may originate from a metabolic malfunction in LECs. Yanshole et al. (2019) conducted a study that demonstrated the significance of metabolite concentrations in the lens and corresponding aqueous humor in terms of lens protection. The study revealed that the lipids crucial for lens protection are synthesized within LECs, suggesting that the development of age-related nuclear cataracts may arise from LEC malfunction. However, the cataractous lenses in the above studies were from the extracapsular cataract extraction surgery. The surgery is relatively traumatic, and its clinical application is gradually decreasing. At present, the phacoemulsification technique for cataract surgery is becoming increasingly popular. The more easily obtainable tissue during surgery is the anterior capsule of the lens, and the LECs under the anterior capsule also have great research value. As far as we know, the utilization of lipidomics in investigating the composition of lipids found in LECs of individuals with ARC has not been documented. Therefore, our study aimed at analyzing the lipids in the LECs of ARC patients with different lens opacity, which could extend our knowledge of the cataract pathological mechanism and explore new methods for the treatment of ARC.

Materials and Methods

Study participants and sample collection

This study enrolled patients diagnosed with ARC and planned to undergo cataract extraction in the ophthalmology department of Shanghai Tongren Hospital between 1 December 2021 and 1 March 2022. All patients had a comprehensive eye examination and signed an informed consent form before surgery. The eye examination includes vision, optometry, intraocular pressure, slit lamp examination, corneal topography, fundus examination and eye B-ultrasound. According to the Lens Opacities Classification System (LOCS) III, patients were classified as having mild ARC if the cortical cataract score was <2, the nuclear opalescence or nuclear color was <2, and the posterior subcapsular cataracts score was <2 (Chylack et al., 1993). Patients with scores greater than 3 for all three measures were classified as having severe ARC.

Exclusion criteria: patients with diabetes, thyroid diseases, cardiovascular and cerebrovascular diseases, and mental disorders were excluded. Finally, 20 patients were included in the study (10 cases in each group).

All operations were performed by the same physician. Phacoemulsification combined with intraocular lens implantation were performed in all patients. During the operation, the continuous circular capsulorhexis technique was used to peel off the anterior LC with a diameter of about 5 mm as the tissue sample, followed by immediate transfer to the Eppendorf tube and −80 °C storage.

All procedures of this study followed the Declaration of Helsinki, with all patients providing written informed consent for sample extraction. The study obtained ethics approval from the Research Ethics Committee of the Tong Ren Hospital, affiliated with Shanghai Jiao Tong University School of Medicine, Shanghai, China (approval no. 2021‑078‑01).

Liquid chromatography‑mass spectrometry profiling and data processing

The samples were taken out of the −80 °C refrigerator carefully and thawed slowly on ice, followed by adding 1.5 ml chloroform/methanol (2:1) solution and then adding 0.5 ml pure water and subjected to 1 min vortex. Subsequently, the mixture went through 10 min centrifugation (1,006.2 ×g, 4 °C), then transferred all organic phases to another test tube, and finally dried with nitrogen gas. After drying, the sample was re-dissolved in 100 μL of isopropanol/methanol (1:1) and added 4 μL of internal standard LPC (0.14 mg/ml). Then the mixture was vortexed and centrifuged for 10 min (16,099.2 ×g, 4 °C). Eventually, the supernatant was carefully transferred to a clean bottle.

The data were analyzed by the liquid chromatography-mass spectrometry (LC-MS) system (Q Active Orbitrap; Thermo Fisher Scientific, Waltham, MA, USA). The column was ACQUITY UPLC BEH C18 (2.1*100 mm 1.7 μm). The chromatographic separation conditions: column temperature: 40 °C; flow rate: 0.3 mL/min; Mobile phase A: Acetonitrile/water (6:4, v/v, with 10 mM Ammonium formate); Mobile phase B: Iso-Propyl alcohol/Acetonitrile (9:1, v/v, with 10 mM Ammonium formate); Injection volume: 1 μL (positive ion), 3 μL (negative ion); automatic injector temperature: 4 °C. The mobile phase gradient elution program is shown in Table 1. The analyses were conducted using both positive (+) and negative (−) ion modes in ESI.

Table 1 The gradient of mobile phase.

Time (min)	Flow rate (mL/min)	A (%)	B (%)	
0.00	0.3	70	30	
10.50	0.3	0	100	
12.50	0.3	0	100	
12.51	0.3	70	30	
16.00	0.3	70	30	

Quality control (QC) samples were made by mixing equal amounts of the test samples, and were tested on the machine before, during, and after LC-MS/MS injection of the test sample. Blank samples were detected during detection and added during data extraction. The peak area value during statistical analysis was the integrated peak area value after deducting the blank.

Data processing: (1) Filter individual Peaks. Only retain peak area data with single group values not exceeding 50% or all group values not exceeding 50%. (2) Simulate missing values in the original data. The numerical simulation method is filled by the minimum half method. (3) Data standardization processing. The peak area normalization method was used: firstly, performed peak area analysis on samples, then calculated the area of each peak, then calculated the total area of all peaks in each sample, and finally calculated the percentage of each peak to the total peak area.

Statistical analyses

Firstly, the original files obtained from LC-MS were imported into Lipid search software (Thermo Fisher Scientific, Waltham, MA, USA) to perform spectral processing and database search. The multivariate analysis was conducted to reveal the differences in lipids among different groups, including principal component analysis (PCA) and orthogonal projections to latent structures-discriminate analysis (OPLS-DA) (Bylesjo et al., 2008; Chong et al., 2018; Pang et al., 2021). The study used SIMCA software (Sartorius Stedim Data Analytics AB, Umea, Sweden) for performing UV formatting of data and then performing automatic modeling and analysis. The criteria used for screening differential lipid lipids were P-value < 0.05 for the student’s t-test, and the variable importance in the projection (VIP) of the first principal component of the OPLS-DA model was >1. Meanwhile, the area under the receiver operating characteristic curve (AUC) for each lipid metabolite was calculated. Further analysis of the metabolic pathways associated with the identified differential lipids was performed using the Kyoto Encyclopedia of Genes and Genomes (KEGG; http://www.kegg.jp/kegg/pathway.html) (Kanehisa et al., 2014). Statistical analysis was performed using version 22.0 Statistical Product and Service Solutions (SPSS, Chicago, IL, USA). A P value of <0.05 was considered statistically significant. Pearson’s chi-squared test was used to distinguish the characteristics of the participants. The Spearman correlation analysis was used to analyse the metabolite content with the patient’s vision, diopter and intraocular pressure. P-values were corrected for multiple comparisons (FDR).

Results

The participant features

A total of 20 patients underwent cataract surgery (10 cases in the mild ARC group, 10 cases in the severe ARC group). The average age of the participant was 70 years (60–79 years) in the mild ARC group and 75.90 years (63–87 years) in the severe ARC group, with no significant difference between the two groups in age (P = 0.095). Females accounted for 80% (8/10) in the mild ARC group and 50% (5/10) in the severe ARC group, with no significant difference between the two groups in gender (P = 0.16).

Metabolic profiles of LECs

The PCA modeling was conducted for obtained datasets revealing that all samples were within the 95% confidence interval (CI) (Hotelling’s T-squared ellipse) (Fig. 1). It showed that the QC samples had good aggregation. Subsequently, an OPLS-DA model was constructed to conduct sample classification for each dataset, revealing a significant difference (95% CI) between the two groups as shown by the score scatter plot of the OPLS-DA model (Fig. 2). The validation plot provided strong support for the validity of the model, as the majority of permuted R2 and Q2 values on the left side of the plot were observed to be lower than the corresponding original points on the right side (Fig. 3) R2Y (cum): 0.935, Q2 (cum): 0.339.

Figure 1 Score scatter plot of PCA model for mild vs. severe ARC groups.

Figure 2 Score scatter plot of OPLS-DA model for mild vs. severe ARC groups.

Figure 3 Permutation test of OPLS-DA model for mild ARC group vs. severe ARC group.

Identification of potential biomarkers

Further screening and analysis of differential lipids were conducted utilizing the criteria of the P-value < 0.05 for the student’s t-test and P-value > 1 for the VIP of the first principal component of the OPLS-DA model. Finally, in LEC lipids, 26 lipids have been identified as potential biomarkers between mild ARC and severe ARC, with AUC values of 0.67–0.94 (Table 2). Subsequently, we visualize the results of the screened differential lipids. The results were presented in a volcano plot (Fig. 4). It shows nine lipids with reduced intensity and 17 with increased intensity. Compared with the mild ARC group, nine lipids were found to be decreased in the severe ARC group, including bis-methyl phosphatidic acid, ceramides, diglyceride, phosphatidylglycerol, and triglyceride series. A total of 17 lipids were increased in the severe ARC group, including ceramides, diglyceride, simple Glc series, methyl phosphatidylcholine, phosphatidylcholine, phosphatidylethanolamine, triglyceride, and wax esters.

Table 2 Potential metabolic biomarkers identified in LECs of ARC patients.

Lipid name	Class	RT	VIP	P-VALUE	LOG_FOLDCHANGE	Up/Down regulation	AUC	
BisMePA(48:5)	BisMePA	11.444	1.535	0.027	−1.359	Down	0.74	
Cer(d20:0/16:0)	Cer	11.202	2.644	0.009	−2.873	Down	0.89	
Cer(d18:0/18:0)	Cer	11.326	2.940	0.001	3.168	Up	0.94	
Cer(d22:1/16:0)	Cer	11.538	2.264	0.042	1.529	Up	0.81	
Cer(d22:0/18:0)	Cer	12.857	2.207	0.006	5.736	Up	0.82	
DG(18:0/16:0)	DG	10.835	2.088	0.005	−3.471	Down	0.79	
DG(18:0/16:0)	DG	10.831	1.785	0.011	−2.175	Down	0.8	
DG(18:0/16:0)	DG	11.08	2.94	0.000	2.381	Up	0.92	
DG(36:2e)	DG	13.114	2.093	0.004	4.144	Up	0.81	
Hex1Cer(d39:3)	Hex1Cer	10.915	2.122	0.002	0.821	Up	0.91	
Hex2Cer(d13:0/19:1)	Hex2Cer	11.201	1.760	0.049	2.308	Up	0.67	
MePC(35:3)	MePC	10.416	1.670	0.036	1.703	Up	0.79	
PC(20:4e/18:0)	PC	11.89	1.607	0.025	2.285	Up	0.82	
PC(16:0/22:6)	PC	10.405	1.112	0.020	1.800	Up	0.78	
PC(18:0/22:6)	PC	11.213	1.868	0.049	1.803	Up	0.75	
PE(40:2)	PE	11.908	2.574	0.000	0.623	Up	0.93	
PG(18:0/9:0)	PG	9.016	2.126	0.032	−1.066	Down	0.79	
TG(12:0e/6:0/16:0)	TG	10.826	2.432	0.004	−2.830	Down	0.87	
TG(12:0e/6:0/16:0)	TG	10.834	1.735	0.022	−2.762	Down	0.7	
TG(12:0e/6:0/16:0)	TG	11.105	3.024	0.000	2.614	Up	0.92	
TG(20:3e/6:0/11:1)	TG	12.917	1.896	0.008	−2.001	Down	0.76	
TG(16:1/14:0/24:0)	TG	12.051	1.906	0.021	0.957	Up	0.76	
TG(16:0/18:1/22:0)	TG	12.238	2.147	0.035	2.215	Up	0.79	
TG(16:1/20:5/20:5)	TG	11.747	2.397	0.010	−1.274	Down	0.83	
WE(13:0/21:2)	WE	11.446	2.735	0.003	5.749	Up	0.94	
PE(18:0/20:4)	PE	11.669	1.593	0.038	2.648	Up	0.71	

Figure 4 Volcano plot for mild ARC group vs. severe ARC group.

Pathway analysis for potential biomarkers

This study involved the utilization of databases such as KEGG and MetaboAnalyst to explore the potential metabolic pathways of lipids (Pang et al., 2021). In order to gain insight into the metabolic pathways associated with these potential biomarkers, additional pathway analysis was performed, encompassing enrichment analysis and topology analysis. Finally, five pathways demonstrated significant enrichment at a significance level of 0.10. These pathways include glycerophospholipid (GPL) metabolism, linoleic acid metabolism, α‑linolenic acid metabolism, glycosylphosphatidylinositol (GPI)‑anchor biosynthesis, and arachidonic acid metabolism (Fig. 5). In particular, the GPL metabolism exhibited significant alterations, indicating the crucial involvement of these metabolic markers in this pathway regulation.

Figure 5 Enrichment analysis and pathway topology analysis for potential metabolic biomarkers.

Correlation between lipids and cataract

The interconnection of metabolic reactions and their regulation in organisms gives rise to the formation of intricate pathways and networks involving various genes and proteins. These components mutually influence and regulate one another, ultimately resulting in systematic alterations within the lipid group. Four potential metabolic biomarkers related to the GPL pathway, DG (18:0/16:0), PC (16:0/22:6), PC (18:0/22:6), and PE (40:2), were selected to analyze the correlation with the clinical symptoms of ARC. Using the Spearman correlation calculation method, we analyzed the metabolite content and the patient’s vision, diopter, intraocular pressure and other conditions for correlation analysis. The results showed that the content of PE (40:2) in the LECs of patients with mild ARC was significantly higher than that of the severe ARC group and was negatively correlated with the LogMAR visual acuity of ARC patients (P = 0.013), while the other three lipids were not correlated with the visual acuity (P > 0.05). This result suggested that PE (40:2) might be a protective active lipid metabolite for ARC (Table 3).

Table 3 Correlation analysis between potential metabolic biomarkers and the best corrected visual.

Lipid name	Class	RT	r	P-VALUE	
DG(18:0/16:0)	DG	10.835	0.426	0.061	
PC(16:0/22:6)	PC	10.405	−0.195	0.409	
PC(18:0/22:6)	PC	11.213	−0.165	0.487	
PE(40:2)	PE	11.908	−0.544	0.013	

Discussion

ARCs are still a common disease in clinical work. As we know, the pathogenesis of cataracts is complex and is the result of multiple factors accumulation. Therefore, further in-depth exploration is needed. Herein, we examined the metabolic differences in the LECs of patients with varying ARC degrees and identified 26 lipids as possible biomarkers in LECs of ARC patients with varying degrees of lens opacity. They could be used to distinguish between mild and severe ARC patients. And the AUC values of these lipids are 0.67–0.94, indicating that these lipids have certain diagnostic value for ARC.

A significant difference was revealed between the metabolomic components of normal transparent and ARC lens (Tsentalovich et al., 2015). The most significant difference is the compounds that contribute to lens cell protection and metabolic activities, which include antioxidants, ultraviolet filters, and osmotic pressure, suggesting that ARC occurrence may be due to the metabolic dysfunction of LECs. To our knowledge, there are currently few studies using human LECs samples to explore lipids related to ARC. Therefore, our study has a certain degree of innovation. Another research showed that disorders in the cholesterol synthesis of LECs could cause lens opacity (Zhao et al., 2015). Lanosterol holds significant importance as an intermediate product within the process of cholesterol synthesis. The stabilization of molecular chaperones and the enhancement of proteasome activity due to lanosterol have been shown to have a positive effect on the solubility of lens protein. This suggests that the lipid metabolism of LECs is involved to a great extent in maintaining lens transparency (Liu et al., 2023). This study unveiled differential expression of bis-methyl phosphatidic acid, ceramides, diglyceride, simple Glc series, phosphatidylcholine, phosphatidylethanolamine, and triglyceride lipids in LECs of patients with mild and severe ARC. This result indicated that lipids might be involved in the pathological and physiological processes of cataracts.

In this study, it was also shown that although the ceramide belong to the same class of lipids, their concentrations vary in different degrees of cataracts. Compared to the mild ARC group, the concentration of Cer (d20:0/16:0) in the severe ARC group decreased, while Cer (d18:0/18:0), Cer (d22:1/16:0), and Cer (d22:0/18:0) increased. This suggested that they may play different roles in the pathological process of cataracts. Previous studies have shown that ceramide is an important bioactive substance in the body, playing a decisive role in mediating the signal transduction process of cell apoptosis, and is one of the important characteristic markers for measuring cell apoptosis (Gomez-Munoz, 1998; Srivastava, Ramana & Bhatnagar, 2005). Therefore, our research findings suggest that our future research can further explore the mechanism of cataract occurrence from the perspective of apoptosis.

We conducted further research by searching public databases such as KEGG and using the MetaboAnalyst platform in an effort to identify potential pathways of lipids of ARC (Kanehisa et al., 2014; Pang et al., 2021). The findings of our study indicate a significant enrichment of the GPL metabolic pathway, suggesting a crucial regulatory role of these metabolic markers in this pathway. GPLs are the most abundant phospholipids in human cells (Yu et al., 2017). In cell physiology, GPLs are involved in the formation of biofilms, as well as in the recognition and signal transduction of proteins by cell membranes. Previous studies have confirmed that lipid peroxidation (LPO) is one of the main factors in the occurrence and development of cataracts (Ahmad & Ahsan, 2020). As is well known, lipid peroxidation concerns all lipids. Therefore, the GPL pathway may also be involved in the process of LPO. In future research, we could study on whether and how the GPL pathway affects LPO to investigate its role in the development of cataracts.

Our study showed that among those lipids, the PE (40:2) metabolite in the LECs of patients with mild ARC was significantly higher than those of patients with severe ARC, indicating that the lipids of PE (40:2) had significant effects in cataracts and classification. PE (40:2) was negatively correlated with the LogMAR visual acuity of patients with ARC (P = 0.013), patients with higher PE (40:2) could have better vision. We speculated that PE (40:2) might be an active lipid metabolite with lens protection function. High concentrations of PE (40:2) might alleviate the severity of cataracts. Our research team is currently conducting in-depth research on this speculation. PE (Phosphatidylethanolamines) widely exists in the biological world and is the second most abundant phospholipid in the eukaryotic cell membrane. As a lipid chaperone, PE has considerable activity, which helps to fold some membrane proteins and contributes significantly to autophagy initiation (Patel & Witt, 2017). The metabolic disorder of PE is related to multiple chronic disorders, including Alzheimer’s, Parkinson’s, and non-alcoholic liver disorders, as well as atherosclerosis, insulin resistance, and obesity (van der Veen et al., 2019). PE also plays an important role in lens metabolism (Huang et al., 2006). Refer to HMDB human metabolite database (https://hmdb.ca/); it is found that PE (40:2) has good lipophilicity and is easy to enter cells through the cell membrane to function. As a small molecule of active lipid, PE (40:2) may participate in preventing and managing early ARC in the future. The identification of new biomarkers of lipids in the LECs of human cataracts provides an understanding of potential new pathogenic pathways for this disease. The finding has translational value for developing new cataract drug treatment methods in the future.

Inevitably, this study had some limitations, one of which was the small sample size. This may affect the expression of some products of lipids. In addition, due to the limitation of sample size, the degree of cataract could not be divided into more detailed groups. In the future, we plan to continue expanding our sample collection and conducting more in-depth research. Since it is difficult to obtain the anterior capsule tissue sample of transparent lens, the blank control group was not set up in this study. In the future, we are planning to obtain some anterior lens capsule samples of transparent lens from patients who need to remove lens during vitrectomy as the control group for more in-depth research and analysis.

Conclusions

In our study, we investigated metabolic markers in human LECs from patients with ARC. The results indicated that 26 lipids had been identified as potential biomarkers in LECs of ARC patients with varying degrees of lens opacity that could distinguish severe ARC from mild ARC. In comparison to the mild cataract group, we found that nine lipids decreased and 17 lipids increased in patients with severe ARC. In the pathway analysis, GPL metabolism was significantly impacted, which might be related to the etiopathogenesis of ARC. PE (40:2) may be an active lipid metabolite that plays a role in protecting the lens and maintaining its transparency. Overall, our study provided a valuable attempt to extend our knowledge on the pathological mechanism of ARC and offered new directions for exploring the prevention, control and treatment of ARC.

Supplemental Information

Supplemental Information 1 Raw data.

We would like to thank all of the participants, as well as our colleagues, who helped in completing this study in Shanghai Tongren hospital.

Additional Information and Declarations

Competing Interests

Author Contributions

Human Ethics

Data Availability

The authors declare that they have no competing interests.

Yingying Gong conceived and designed the experiments, performed the experiments, analyzed the data, prepared figures and/or tables, authored or reviewed drafts of the article, and approved the final draft.

Qingquan Wei performed the experiments, analyzed the data, prepared figures and/or tables, and approved the final draft.

Liying Luo performed the experiments, prepared figures and/or tables, and approved the final draft.

Wei Qiu performed the experiments, prepared figures and/or tables, and approved the final draft.

Yanyun Jiang conceived and designed the experiments, authored or reviewed drafts of the article, and approved the final draft.

The following information was supplied relating to ethical approvals (i.e., approving body and any reference numbers):

Research Ethics Committee of the Tong Ren Hospital affiliated with Shanghai Jiao Tong University School of Medicine, Shanghai, China

The following information was supplied regarding data availability:

The raw measurements are available in the Supplemental Files.

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
