# Peer review of "A lipidomic study on the lens epithelial cells of patients with age related cataracts"

_PeerJ, doi:10.7717/peerj.17998_

## Round 0.1 · original submission · Major Revisions

It has been reviewed by two experts in the field. Revisions are
necessary before the manuscript is suitable for publication.

Reviewer 1 ·

Basic reporting

Authors performed an interesting analysis on the anterior capsule and adherent subcapsular epithelium (LEC) specimen using high-resolution LC-MS technique. As a result, they putatively identified several aspects that correlate with the two types of cataract maturity. English is quite decent. The text in general is readable.

Experimental design

The design of the experiment seem to be adequate, as well as the analysis techniques used.

Validity of the findings

The narrative and main conclusions of the manuscript are too bold, and should be rewritten. The details are provided in the “General comments” section.

Additional comments

Dear Authors,

Here is my opinion about the manuscript entitled "A metabolomic study on the lens epithelial cells of patients with age related cataracts" by Yingying Gong, Qingquan Wei, Liying Luo, Wei Qiu, Yanyun Jiang, sent to PeerJ.

Authors performed an interesting analysis on the anterior capsule and adherent subcapsular epithelium (LEC) specimen using high-resolution LC-MS technique. As a result, they putatively identified several aspects that correlate with the two types of cataract maturity. The design of the experiment seem to be adequate, as well as the analysis techniques used. English is quite decent. Although there are several points that need to be addressed. The main conclusions of the manuscript are too bold, and should be rewritten.

Major.
• First, authors examine only lipids. Therefore, the entire narrative in the manuscript including title should be about lipids and lipidomics, rather than the wider field of metabolites and metabolomics.
• What is the procedure for the lipid identification? What is the compound identification level, according to Metabolomics Standards Initiative (MSI) guidelines (Sumner et al., 2007)? Were substances subjected to MS2 or chemical standards examination? There is a feeling, that probably lipids were identified only based on their exact masses. Why, for example, compound 31 and compound 32 from raw_data file with m/z 568.56632 had been identified as different Cer(d20:0/16:0) and Cer(d18:0/18:0) lipids? The same question is for all other compounds in this file (with the same m/z), how their identity (and names) were confirmed (BisMeLPA(10:0e) vs MG(18:2e), MG(16:0) vs DG(16:0e), etc.)?
• Performing of a pathway analysis based on lipid compounds yielded enrichment of a most abundant lipid GPL pathway. That is quite obvious. Later speculating on the identified this way lipid pathway seem to be too bold (lines 267-283, etc.). No proof of association of solely GPL pathway with mentioned in ARC stresses in Ref (Ahmad, 2020). Although indeed obviously well-known lipid peroxidation concerns all lipids (Ahmad, 2020). This part this part needs to be completely rewritten.
• Correlation between metabolites and cataract maturity part. Lines 225-238 and later in discussion and conclusions. Authors narrowed study to only 4 lipids from the GPL pathway. It is better to examine correlation of all detected compounds with the cataract maturity, rather than artificially narrow their number. Here, again one putatively annotated compound PE(40:2) and way too bold and unfounded conclusions from it: “significant significance in cataracts” and “participate in maintaining lens transparency”. HMDB yields 44 compounds for this single m/z of 800.61638 (M+H and M+NH4), none of them is PE(40:2). Again speculating on the identified this way lipid pathway seem to be too bold (lines 267-283; 302-303, etc).
Could authors provide HMDB compound ID for PE(40:2) where authors get its properties?
Moreover PE is a cell membrane constituent. Phrase in line 301 “PE … enter cells through the cell membrane to function” is questionable.
• How correlation with cataract stage was performed? No numerical data provided for the examined parameters: “patient’s vision, diopter, intraocular pressure and other conditions” and “LogMAR visual acuity”. Was it parametric Pearson test or non-parametric Spearman test? (There is no Sperson test). What are “r” and “p-value” in the Table 2? That is the interesting part, and the data analysis here should be performed conscientiously.
• Line 281 and other parts of the manuscript. Authors compare only two cataract stages without investigation of normal tissue. Phrases such as “correlation … occurrence of cataracts” are incorrect. Better to use “maturity” or “stage” instead of “occurrence”.

Minor.
• Hierarchical clustering heatmap is excessive and unnecessary. All the red and blue compounds are presented in Table 1 as Up- and Down-regulated. Remove or transfer to SI.
• By the way, Table 1 contains only 19 of 22 differential lipids.
• One of PLS-DA or oPLS-DA is excessive and unnecessary. They yield similar results in your case. Remove or transfer to SI.
• How pathway analysis was performed (with what software/methods)? How PubChem database participates in the pathway analysis?
• Lines 97-111. Investigation of Ref (Yanshole 2019) is stated to be the follow-up and re-examination of the work by Ref (Tsentalovich 2015). Thus, rise of most metabolite concentrations in the normal lens in Ref (Tsentalovich 2015) should be attributed to post-mortem processes in cadaveric eyes. Although several particular metabolite differences indeed occur due to cataract processes. In both works, cataract samples were obtained post-operationally from living patients. This part and especially statement in lines 109-111 should be corrected.
• Line 148 what is all organic phases? Did you combine water-methanol and chloroform-methanol fractions?
• Line 150 what is LPC?
• Line 152 Q Active -> Q-Exactive
• Describe LC parameters
• For the multivariate data analysis did you perform scaling or normalization?
• Line 165 What is UV formatting?
• Line 165 What is automatic modeling?
• Line 184-185. What conclusion comes from PCA? What conclusion comes from that all samples are being inside the CI 95%?
• Lines186-193. There are no CI for groups separately shown. Thus, authors cannot state about “significant difference”. Moreover, supervised analyses in principle cannot state significant differences. If the separation was observed on unsupervised PCA, than this could be.
• Line 163 Ref (Chong 2018) does not describe oPLS-DA method, but an older version of the well-known MetaboAnalyst platform.
• Tables 1 and 2 are not tables, but pictures/screenshots.
• Line 43: “It is estimated that cataract patients will reach 151.14 million in China in 2025” without reference, while Ref (Liu 2017) states: “An estimated 95 million people worldwide are affected by cataract.” Describe this discrepancy.
• All references should be checked. Examples: Blindness et al – there is no author with the surname “Blindness”; “Zhenzhen Liu” is not a surname, but a full name.
• Line 53 LC abbreviation should be disclosed.
• Sentence in lines 80-84 is inconsistent.
• Line 94. In Ref (Lains 2019) authors studied also urine.
• Line 247. Terminology: metabonomic -> metabolomic. Although both can be used, unification favors reading.
• Lines 94 and 109. Terminology: choose either intraocular fluid or aqueous humor. Although both can be used, unification favors reading.
• Correct ACR to ARC in all cases.
• Lines 255-258. Why the description of lanosterol is needed here?
• Line 292 citation needed.
• Why parametric Student’s t-test was chosen as the method for calculating p-values (instead of Mann-Whitney)? Have the values been checked for the normality of the distribution? Were p-values then corrected for multiple comparisons (FDR)?
• There is an outlier in PLS-DA and oPLS-DA plots. Was it examined?

Reviewer 2 ·

Basic reporting

The article submitted for review concerns metabolomics performed on lens epithelial cell samples from patients with age-related cataracts. The article is written in correct English, although the authors made some minor mistakes. The introduction describes the research problem sufficiently, and the literature has been selected appropriately. The structure of the article is appropriate. The figures presented are appropriate in the context of the article, although their descriptions and/or captions should be corrected. The authors provided raw data, but only part of it is in English. The reviewer is unable to read the entire file.
Detailed questions are included in the additional comments.

Experimental design

The purpose of the research has been clearly defined and is in line with the current research trend. The authors provided the appropriate consent necessary for research using human material. The technology used seems mostly correct. However, the research method as well as the data analysis and statistics were described insufficiently both for a reliable assessment by the reviewer and replication of the experiment by other researchers.
Detailed questions are included in the additional comments.

Validity of the findings

The reviewer has doubts about the data analysis, quality control, and statistics. Detailed questions are included in the additional comments.
The conclusions answer the research question. They summarize well the part of the results that were included in the discussion.

Additional comments

The reviewer has questions as well as some suggestions for improving the quality of the manuscript.

General note: the authors talk about “biomarkers” in the abstract, but in the discussion, they talk about “potential biomarkers”. Please unify this.

Introduction
1. The first paragraph of the introduction describes cataracts with references to the situation in the Chinese population. This information is interwoven with each other. The reviewer suggests separating this information to improve the readability of the information presented: first describe the cataract and then mention the Chinese environment.
2. Line 40: Citation missing.
3. Line 48: Abbreviation expansion is missing.
4. Some sentences are too long and difficult to understand. The reviewer suggests rewriting them: lines 72-75, 79-81 and 89-93.

Materials and methods
5. Can the authors explain why mental disorders were an exclusion criterion?
6. The authors do not mention QC samples throughout the manuscript, but they are included in the raw data. How were the QC samples prepared and used? Please add this information to your manuscript.
7. Did the authors make and use an extraction blank in the data analysis? If so, please add this information to your manuscript. If not, explain why.
8. What were the parameters of the method used to generate the data? What column was used for chromatographic separation? What phases were used? Please add this information to the manuscript or please provide a citation to a publication that already contains this information.
9. What were the parameters used to process the data? Please add this information to your manuscript.
10. Please provide the versions of the software used for statistical analyses.
11. It is common to use Pareto formatting on log transformation data for multivariate statistics. Can the authors explain why they decided to use UV formatting?
12. Were there any missing values in the generated data? If so, how were they replaced?
13. What software was used to perform enrichment analysis and pathway topology analysis? How were individual AUCs calculated? Please add this information to your manuscript.

Results:
14. Were patients in both groups compared only by age and gender? Could the authors provide additional information, e.g. patients' BMI, type of cataract (cortical, nuclear, subcapsular), medications taken, or parameters later used for correlation? The reviewer suggests collecting and presenting all parameters in a table.
15. Please add QC samples to the PCA model and refer to them in the results or discussion.
16. Why did the authors decide to show both PSL-DA and OPLS-DA models?
17. Please add the values of Ry2 and Q2 to the captions under Figures 2 and 3.
18. The reviewer suggests adding volcano plots to the presented PLS-DA and OPLS-DA models, at least as panels B. This will increase the transparency of the analyses performed and the presented results.
19. Was any correction of the obtained p-values made? If so, please add this information to your manuscript. If not, please explain why.
20. How was the identification of metabolites performed?
21. Table 1 does not include all statistically significant metabolites. Please complete or explain why some metabolites are missing.
22. Please complete the caption under Figure 5. The reviewer guesses that the letters M and S correspond to mild and severe ARC samples.
23. On what basis were metabolites selected for correlation with clinical parameters?
24. Line 208: Please correct the name of the correlation method.

Discussion
25. Lines 223-227: This is a repetition of the sentences from the introduction (lines 89-93). The reviewer believes that there is no need for this. Please remove this fragment. The reviewer leaves it to the authors to decide which fragment should be removed.
26. Line 257: "significant significance" please paraphrase that
27. Lines 267-270: Why did the authors include such detailed information about PE (40:2), such as formula and molecular weight? This information, in the reviewer's opinion, is not important in the context of the discussion.
28. The authors did not discuss all the results in the discussion. For example, ceramides have been completely omitted. They also do not mention the AUC curves, the values of which are given in Table 1. Please add appropriate comments to the discussion.

---

## Round 0.2 · Major Revisions

Revisions are necessary before the manuscript is suitable for publication.

Reviewer 1 ·

Basic reporting

The manuscript was improved.
Triple check references. For example, there is no person with the surname "Collaborators" (Line 45). Triple check "ARC", I've still met ACR in the text.
The MetaboAnalyst is not a database, but rather a platform or a tool.

Experimental design

improved

Validity of the findings

improved

Reviewer 2 ·

Basic reporting

First of all, the submitted versions of the manuscript (with and without track changes) differ from each other. The track changes version does not contain figures or tables but does contain additional content.
Additionally, the authors took into account only some of the reviewer's comments and suggestions. Although the authors wrote in their responses to the reviews that they had added the information, this information is missing in the submitted versions of the manuscript.
Detailed questions are included in the additional comments.

Experimental design

The completed methods description still does not contain all the necessary information and is insufficient to repeat the experiment.
Detailed questions are included in the additional comments.

Validity of the findings

Again, some of the reviewer's suggestions were not taken into account, which does not allow for a full assessment of the research conducted.
Detailed questions are included in the additional comments.

Additional comments

The reviewer will again refer to the fragments that should be corrected for the proper reception of the manuscript.

Introduction:
1. The authors, following the reviewer's suggestion, have rewritten part of the introduction. However, the citation is still missing (lines 43-47).
2. Line 50: The abbreviation expansion is still missing.
3. Changing the order of the sentences did not change their perception. The reviewer suggests a few shorter sentences on lines 74-79 that will convey the information more simply.

Materials and methods:
4. Information about QC samples and extraction blanks should be included in the materials and methods section.
5. Please provide centrifugation parameters in g, not rpm.
6. Why was LPC used as IS? Typically, compounds that do not occur naturally in the organism are used as IS, e.g. drugs withdrawn from use or labeled forms of commonly occurring compounds. Was the used LPC labeled?
7. The authors have added some of the information regarding analysis parameters, although some of the information is included only in the track changes version. There is still no information regarding the phases and gradients used.
8. The authors mention the PLS-DA model, which was ultimately removed. This information should also be removed (line 150).
9. The manuscript still lacks information about the parameters used to process the data and the method of replacing missing values.

Results:
10. The authors, at the reviewer's suggestion, added QC samples to the PCA model. However, the reviewer notes that he was too vague in his request. The reviewer asks authors to mark only QC (the rest of the samples are not grouped) and pay attention to the fact whether the QC samples will be clustering at one point, which would indicate the good quality of the generated data. Given the current figure, the QC samples show a trend based on which additional normalization of the obtained data should be considered.
The authors did not provide any of the above information in the manuscript.
11. Please correct the figure numbers. Figure 5 (line 196) probably refers to the deleted heatmap, while Figure 6 (lines 205-206) is not included. Additionally, there are still no Ry2 and Q2 values for the presented models. And there are no figures and tables in the track changers version manuscript.
12. The presented volcano plot (Figure 4) shows 9 metabolites with reduced intensity and 17 with increased intensity, which is inconsistent with the described results (7 down and 15 up). Is the correct model presented? Additionally, the reviewer suggests assigning the names of the metabolites to the appropriate points on the model.
13. Information about any correction of the obtained p-values is still lacking in the manuscript.

Discussion:
14. The authors did not add to the discussion any other statistically significant metabolites (e.g. ceramides). If the obtained metabolites have not been previously associated with similar research results, it would be even more important to mention them.

---

## Round 0.3 · Major Revisions

Revisions are still necessary before the manuscript is suitable for publication

Reviewer 1 ·

Basic reporting

done

Experimental design

done

Validity of the findings

done

Additional comments

Terminology. Authors examine only lipids, but still wider term "metabolites" is extensively used. Please replace "metbolites|metabolomics" and corresponding wordforms by "lipids|lipidomics" and corresponding wordforms throughout the whole manuscript. For example:
A lipid metabolomic -> A lipidomic
In LEC metabolites, 22 metabolites -> In LEC lipids, 22 lipids
potential metabolic biomarkers -> potential lipid biomarkers
Metabolic profiles -> Lipid profiles

line 172. Write in explicit form what test was used to calculate p-value. Write in explicit form, whether the p-value was corrected to multiple comparisons (FDR).

Line 198-211. It is not entirely clear, why several lipids were excluded based on the results from volcano plot. There is a feeling that authors simply adjust results to previous findings. The results given by volcano is more correct from mathematical viewpoint, and have more statistical and scientific soundness. I would suggest to depart from the volcano, rather than from the heatmap.

Reviewer 2 ·

Basic reporting

The authors took into account most of the reviewer's comments and suggestions. Although based on the authors answers, the reviewer is concerned if the authors understand fully what they doing.
Detailed questions are included in the additional comments.

Experimental design

The method description is improved but still does not contain all the necessary information. Detailed questions are included in the additional comments.

Validity of the findings

Again, some of the reviewer's suggestions were not taken into account, which does not allow for a full assessment of the research conducted.
Detailed questions are included in the additional comments.

Additional comments

The reviewer will again refer to the fragments that should be corrected for the proper reception of the manuscript.

Minor things:
1. The abbreviation expansion should be performed in the abstract section as well as the first time appearing in the main text. So line 53 is still missing.
2. The authors have added information regarding the gradients used. However, there is still no information regarding the phases.
3. Line 161: the comma is unnecessary.
4. Line 261: information about carnitines should be provided in a separate paragraph.

Major concerns (please answer the “Q” sentences):
The authors added QC samples to the PCA model and marked QC, but still marked the rest samples as "other" instead of "no class". Presenting data in that way causes it is still impossible to say anything about the quality of the data. Also, authors write what they normalize the data using the total ion current (TIC) of each sample. Q: Could the authors explain what this type of normalization is and how they did this? Based on the plot presented by the authors in the reviewer response, the QC samples are separate from each other and show a disturbing trend. If this is the data shown after normalization, it means that the wrong normalization was used or even data treatment was inappropriate. Such data should not be used for statistical analyses. The reviewer suggests authors check everything once again. Q: Please clearly demonstrate that the data was of adequate quality, both by presenting the QC samples on the PCA model and by an appropriate description in the manuscript. Additionally, showing the PCA model divided into groups is not informative and does not increase the value of the work in any way - the model is not of good quality.
In the case of the OPLS-DA model: the reviewer asked for the Ry2 and Q2 values, to which the authors replied that they were displayed in the model. The reviewer suspects that the authors do not understand what he is asking for. The parameters given on the model usually should be removed from the figures. The values the reviewer talks about are the parameters that can be read from the Simca program (marked with a red frame in the attached figure as an example), which are the real parameters based on which the quality of the obtained models is determined. Q: So please, provide the actual Ry2 and Q2 values.
However, the reviewer's greatest concern is the volcano plot. The authors did not answer the reviewer's question about the differences in the number of metabolites between the volcano plot and the rest of the results. Such inaccuracies should not occur. The results on the volcano plot should be 100% consistent with the results presented in Table 2. The differences should not be explained by obtaining lipids with similar functions or from the same pathway. Additionally, the authors took the opportunity to re-add the heatmap to the manuscript, which Reviewer 1 clearly noted did not add any value during the first round of review. Q: Therefore, the reviewer again asks the authors to check whether the presented model was properly prepared and whether it refers to the presented results.

Annotated reviews are not available for download in order to protect the identity of reviewers who chose to remain anonymous.

---

## Round 0.4 · Minor Revisions

The authors have addressed the reviewer’s suggestions. There is only one thing remaining that needs to be included in the manuscript. Please explain in detail the peak area normalization method and demonstrate that the data was of adequate quality. Reviewer 2 suggested to present the QC samples on the PCA model and to include an appropriate description in the manuscript.

---

## Round 0.5 · accepted · Accept

I have now had the opportunity to read your revised manuscript, and your responses to the comments. I believe that you have addressed the concerns raised, and I am happy to accept your manuscript.